# Flexible, Multifunctional Micro-Sensor Applied to Internal Measurement and Diagnosis of Vanadium Flow Battery

**DOI:** 10.3390/mi13081193

**Published:** 2022-07-28

**Authors:** Chi-Yuan Lee, Chia-Hung Chen, Chin-Lung Hsieh, Chong-An Jiang, Siao-Yu Chen

**Affiliations:** 1Yuan Ze Fuel Cell Center, Department of Mechanical Engineering, Yuan Ze University, Taoyuan 32003, Taiwan; s102080202@gmail.com (C.-A.J.); s1080823@mail.yzu.edu.tw (S.-Y.C.); 2HOMYTECH Global Co., Ltd., Taoyuan 33464, Taiwan; chenjahon@gmail.com; 3Institute of Nuclear Energy Research, Taoyuan 325207, Taiwan; clhsieh@iner.gov.tw

**Keywords:** vanadium redox flow battery, MEMS, flexible multifunctional micro-sensor, temperature distribution, monitor/control, real-time micro-diagnosis

## Abstract

The vanadium redox flow battery (VRFB) system is an emerging energy storage technology with many advantages, such as high efficiency, long life, and high safety. However, during the power-generation process, if local high temperature is generated, the rate of ions passing through the membrane will increase. In addition, it will also cause vanadium pentoxide molecules (V_2_O_5_) to exist in the solid state. Once the solid is formed, it will affect the flow of the vanadium electrolyte, which will eventually cause the temperature of the VRFB to continue to rise. According to the various physical parameters of VRFB shown in the literature, they have a significant impact on the efficiency and life of VRFB. Therefore, this research proposes to develop flexible multifunction (voltage, current, temperature, and flow) micro-sensors using micro-electro-mechanical systems (MEMS) technology to meet the need for real-time micro-diagnosis in the VRFB. The device is embedded in the VRFB of real-time microscopic sensing and diagnosis. Its technical advantages are: (1) it can simultaneously locally measure four physical quantities of voltage, current, temperature, and flow; (2) due to its mall size it can be accurately embedded; (3) the high accuracy and sensitivity provides it with a fast response time; and (4) it possesses extreme environment resistance.

## 1. Introduction

Currently, global power generation relies heavily on fossil-fuel-based energy sources, such as coal, natural gas, and liquid fuels. There are two main problems with using these energy sources: fossil fuels will eventually run out and the emission of greenhouse gas (GHG) and other pollutants can adversely affect ecosystems. Advanced countries have started to formulate environmental protection regulations to reduce greenhouse gas emissions with specific goals. However, since most renewable energy sources are intermittent in nature, it would be more efficient, stable, and reliable to funnel them into the existing grid. By integrating the energy-storage system (ESS) into the power grid, the problem of intermittent power supplies can be completely solved [1,2,3].

ESSs may be divided into five main categories, such as chemical, electrochemical, electrical, mechanical, and thermal energy storage [4]. The most recent and widely available energy storage system is the pumped hydro energy storage (PHES) technology. PHES plants have a broad development prospect. Being the core equipment of PHES plants, pump turbines must frequently change operating conditions to maintain the stability of the power grid. The conversion process is a transient process. During this period, high-amplitude pressure pulsation, which threatens the safe and stable operation, will appear in the vaneless space, inlet pipe, draft tube, and other components of the unit [5].

Although the PHES technology has paired well with energy storage in the preceding years, some of the existing PHES facilities were built to perform large baseload generation and not the irregular, sporadic generation demanded by renewable resource power generation. Large amounts of land are required to build the sizeable infrastructure associated with PHES. PHES systems pose environmental issues and high construction costs amidst the large infrastructure footprint and technical construction. However, the redox flow battery (RFB) hopes to achieve these requirements. A key advantage to redox flow batteries is the independence of energy capacity and power generation. The capacity of the battery is related to the amount of stored electrolyte in the battery system, the concentration of active species, the voltage of each cell, and the number of stacks present in the battery. Conversely, the power generated is related to the behavior of the active species and the electrode size. This facilitates the vast scalability and flexibility of the technology. These are relatively simple systems with few moving parts and often require little operator input, making them low maintenance with little attention needed once set up and running. The combination of all these properties allows the battery to have relatively low running and capital costs, especially compared to other emerging energy-storage technologies [6].

In 2012, the United States began formulating energy-storage-technology development plans. Noteworthily, flow batteries are preferred as the first option. The VRFB has received great attention in ESS applications, demonstrating its flexible design, high efficiency, and long service life. The schematic is shown in Figure 1. When the VRFB energy-storage system operates at room temperature and normal pressure, the heat generated by the battery system can be effectively discharged through the electrolyte and then released outside the system through heat exchange. The electrolyte has high safety, given its nonflammable and nonexplosive characteristics. In addition, the VRFB can discharge at high current density, has a long cycle life, a fast-charging capability, and a wide range of applications. It is also relatively easy to use compared to other energy-storage devices, making it suitable for large-scale use [7,8,9,10,11].

The VRFB energy-storage system uses the redox reaction of the battery’s positive and negative electrolytes to generate the phenomenon of charge and discharge, where energy conversion is based on a reversible electrochemical reaction of two redox pairs. This type of battery consists of two main parts connected by a pump: a stacked battery where the electrochemical reaction occurs and an external tank where the electrolyte is stored [12]. Among the existing flow battery technologies, VRFB is the most promising for large-scale energy storage [13]. Vanadium ions have four valence states of positive two, three, four, and five, which can exist stably. The chemical energy is converted into electrical energy through the redox process of the active material through the circulating flow of the electrolyte. The structure of the VRFB is mainly composed of a proton exchange membrane/separator, two carbon felts, and two bipolar plates. During discharge, the catholyte and the anolyte are pressurized by the pump and then enter the flow channel of the VRFB for the electrochemical reaction to release electric energy. More electrical energy is converted into chemical energy stored in the electrolyte during charging. After the reaction, the liquid flows back to the storage tank again, and the continuous circulation of the electrolyte can complete the charging and discharging process [12,13,14,15].

The vanadium electrolyte of the VRFB undergoes a redox reaction during the charging and discharging process, releasing waste heat and causing the temperature of the vanadium electrolyte to increase. In terms of the temperature difference in the flow channel, the downstream and upstream temperature differences decreased with the increase of the reaction time. This finding may be explained in that the flow channel was farther from the intermediate reaction zone and closer to the bipolar plate without a flow channel, which had good heat dissipation, fewer reactants in the later stage, and significantly reduced heat release. As a result, the temperature difference gradually decreased. Noteworthily, the vanadium electrolyte only takes away excess heat; thus, only a small amount of heat is transferred through the bipolar plates [16].

When the VRFB is charged, the initial flow rate of the channel is slightly higher than the average flow rate, possibly caused by the pressure drop of the fluid. On the other hand, when the vanadium electrolyte flows through the channel, friction occurs between the electrolyte and the tube wall. The friction reduces the propulsive energy of the electrolyte, and the local electrolyte flow within the VRFB decreases and gradually stabilizes [17].

The literature [18,19] shows that if the VRFB generates a local high temperature, it will lead to the precipitation of the electrolyte. Once precipitated in the flow channel, it will seriously affect the flow of the vanadium electrolyte and even block the flow channel. While lower vanadium concentrations and higher sulfuric acid concentrations increase the precipitation temperature. The VRFB system suffers from the problem of electrolyte imbalance, mainly due to asymmetric water crossover from charge and discharge operations. Surface functionalization of ion-exchange membranes is one of the new ways of PWT control. Different methods for suppressing PWT can be proposed, based on capacity fade minimization, among which employ a VFB with a combination of anion exchange membrane (AEM) and cation exchange membrane (CEM); the application of a hydraulic shunt connects the two tanks permanently [20].

One approach to continuously balance the electrolyte is by implementing a regeneration cell to reduce the concentration of VO^2+^ ions in the positive electrolyte tank. However, these side reactions can be minimized by good battery design and voltage control during the operation. For instance, gassing side reactions can be minimized by controlling the voltage during charging and by applying an appropriate current with an adequate electrolyte flow rate [21].

The differential rates of vanadium ion moving across the membrane leads to accumulation of vanadium ions in one half-cell and dilution in the other half-cell. This causes state of charge (SOC) imbalances between the positive and negative half-cells, leading to system capacity loss. The rate of capacity decay is largely dependent on the type of membrane and the permeability rate of the vanadium ions. The loss of capacity due to vanadium ions crossover can be restored by remixing the electrolytes of both sides [21]. A SOC estimation method is proposed, which is based on a measurement of the cell current and an in situ measurement of both half-cell Nernst potentials. A combination of the Coulomb counting and Nernst potential SOC estimation method and deployment of a three-parameter fit allow us to estimate the half-cell’s SOC without any additional ex situ sensor calibration. Instead, reference SOCs, which are typically required for half-cell SOC estimations, can be generated during the operation of the battery system, and costly re-calibrations during the long-term operation can be obsolete [22].

Therefore, controlling the physical parameters of the VRFB system is the key factor to keeping the VRFB system at the best efficiency under the long-term operation. However, due to the severe electrochemical environment inside the VRFB, the general commercial sensor cannot be used for a long time. Thus, it is impossible to obtain the best-operating parameters of the VRFB system. Problems cannot be solved by immediately controlling these system parameters. Therefore, this research will use MEMS technology to design and develop a flexible, multifunctional micro-sensor. According to the real-time micro-diagnosis in VRFB, this research uses MEMS technology to design a flexible multifunction (voltage, current, temperature, and flow) micro-sensor. It embeds inside a VRFB for instant microscopic sensing and diagnosis. Its technical advantages are (1) small size, accurate embedding, unlimited measurement position; (2) high precision, high-sensing sensitivity, and fast-response speed; (3) customizable design and production; (4) multiple data can be detected and collected at the same time (multi-information syn-detection, MIsD); and (5) corrosion resistance. The flexible and multifunctional micro-sensors can accurately measure the voltage, current, temperature, and flow of the VRFB in real time, thus improving the design and performance of the VRFB.

## 2. Process of Flexible, Multifunctional Micro-Sensor

The flexible, multifunctional micro-sensor needs to be embedded in the flow channel of the VRFB, and its electrolyte, sulfuric acid, is a strong acid. Thus, the polymer material, polyimide (PI), with corrosion resistance and electrochemical reaction resistance, was selected as the substrate of the flexible, multifunctional micro-sensor, used to increase the durability of the micro-sensor. Therefore, this study used a polyimide film with a thickness of 50 μm to facilitate embedding inside the VRFB, which reduces the chances of electrolyte leakage.

The flexible substrate should be made of materials with acid resistance, thermal-cycle-aging resistance, and strong electrochemical environment resistance. In addition, it should also not be easily thermally deformed at high temperatures for a long time to reduce the possibility of damaging the micro-sensor. Further, surface micro-machining in the MEMS process was used to avoid damaging the substrate [23], including deposition, lithography, and wet etching, formed by stacking multiple layers of thin films. The production process is shown in Figure 2. The voltage-sensing area is 600 μm × 600 μm, the current-sensing area is 600 μm × 600 μm, the temperature-sensing area is 750 μm × 600 μm, and the flow sensing area is 750 μm × 600 μm.

The detailed steps of the process are as follows:After cleaning the polyimide film with organic solvent ethanol, it was immersed in the boiling organic solvent, acetone. Then, a nitrogen gun was used to remove the highly volatile organic solvent, acetone. Finally, the film was placed into the electron beam evaporation machine.An electron beam evaporation was used to depose chromium and gold thin films. The electron beam evaporator (EBS-500, Junsun technologies Co., Taiwan) used in this study has fast-deposition speed and high-single throughput. The evaporation rate was 0.1 Å/s in the whole process of evaporation. In the process, a 100 Å thick chrome was first deposited as the adhesive layer between the substrate and gold to ensure the compactness of the coating. Then, the 1000 Å gold was deposited as the material for the main sensing structure of the flexible, multifunctional micro-sensor.In the photolithography process, a spin coater was used to uniformly coat a positive photoresist (AZ^®^ P4620, Microchemicals GmbH, Ulm, Germany) on the test piece. Then, the photomask of the flexible, multifunctional micro-sensor was placed in a mask aligner and exposure system (AG-200-4N-D-SM, M&R Nano Technology Co., Taiwan). Finally, the test piece in the developing solution was soaked. This way, the part irradiated by ultraviolet light would disappear so that the pattern could be transferred to the coated substrate.

It was coated on a chrome/gold film at incremental speeds of 500, 1500, and 3000 rpm. Then, the test piece was placed on the hot plate and soft baked at 90 °C for 4 min to remove the solvent in the photoresist. Further, it was exposed with a double-side mask aligner. After the camera takes an 8-s exposure, the test piece can be placed in a developer (AZ^®^ 400 K, Microchemicals GmbH, Ulm, Germany) to begin developing. The concentration of the developer is related to the speed of development. The higher the concentration, the faster the development speed. On the contrary, the lower the concentration, the slower the development speed. However, if the development speed is too fast, the definition pattern will be too visible (the line width is too thin), which may break in the subsequent wet-etching process. Meanwhile, if the development speed is too slow, there will be photoresist residues on the test piece. Therefore, in this experiment, the volume concentration ratio of AZ^®^ 400 K to deionized water is 1:4.5, and the development time is about 30 s.

d.After the pattern was transferred to the positive photoresist (AZ^®^ P4620), the pattern was again transferred to the metal film of chromium (Cr) and gold (Au) by wet etching. The purpose of this step is to define the sensing pattern of the flexible, multifunctional micro-sensor. The etching solutions used in this study are Type-TFA gold-etching solution and Cr-7T chromium-etching solution. Special attention should be paid to time control in the wet-etching process. There will be side etching, considering that wet etching is isotropic. The side-etching phenomenon becomes more serious as the etching thickness increases, which may cause the micro-sensor’s structure to break; hence, it must be handled with care. After the wet etching is completed, the photoresist on the structure was removed from the acetone and methanol.e.To prevent the flexible, multifunctional micro-sensor from being damaged by the locking pressure in the endplate when embedded in the VRFB, a protective layer with high-mechanical strength and resistance to the electrochemical reaction environment must be selected. The electrolyte of the VRFB is a strong oxidizing sulfuric acid solution. Thus, if the protective layer is peeled off, it is easy to accelerate the electrochemical reaction due to electrification, which will accelerate the redox reaction of the metal and destroy the flexible, multifunctional micro-sensor structure. Therefore, selecting the protective layer materials is extremely important. In particular, materials that are insulating, tough, wear-resistant, acid-resistant, and with certain temperature resistance and resistance to electrochemical reactions must be selected. In addition, the protective layer also has an insulating effect, which can avoid the occurrence of a short circuit caused by the flexible, multifunction micro-sensor contacting with the graphite plate. Therefore, the protective layer must be made of an insulating material that can define a pattern. The main purposes of these measures are to expose the sensor head of the micro-voltage and current sensor and directly contact with the flow channel ribs and expose the signal pin (Pad) so that the subsequent signal and output are transferred to the measurement terminal.

In this study, polyimide (Fujifilm Durimide^®^ PI 7320, FUJIFILM Electronic Materials Taiwan Co., Ltd., Hsin-Chu, Taiwan) was selected as the protective layer of the flexible, multifunctional micro-sensor. Since the viscosity of Fujifilm Durimide^®^ PI 7320 is higher than that of AZ^®^ P4620, the spin-coating parameters were 800, 2000, and 2500 rpm, and the thickness was controlled at 6~7 µm. Further, the test piece was placed on a hot plate and soft baked at 90 °C for 8 min. Subsequently, it was exposed with a double-side mask aligner, soaked, and developed in a developer solution.

The development of Fujifilm Durimide^®^ PI 7320 is divided into four steps. First, the more corrosive HTR D-2 is used for 4 min of development. Then, the mixed buffer of HTR D-2 and RER-600 is soaked for 6 min. Further, it is immersed in RER-600 for 6 min to define the pattern. Finally, the test piece is placed in deionized water for 5 min, and the image is completed.

After the development is completed, a hard bake should be performed to complete the curing of the protective layer. In this way, the flexible, multifunctional micro-sensor used in a severe electrochemical environment can have a longer life. The hard baking was carried out at 100 °C, 200 °C, and 300 °C for 30 min and then slowly lowered to room temperature. Figure 3 shows a partial optical microscope view of the flexible, multifunction micro-sensor.

## 3. Sensing Principle of Flexible, Multifunctional Micro-Sensor

### 3.1. Micro Voltage Sensor

The structure of the micro-voltage sensor belongs to the hot-wire type, which is an electrode structure of a thin wire, as shown in Figure 4. Its principle is to use two thin wires to contact both ends of the area to be measured, and the front end exposes the required sensing area as a signal-capture terminal. The rest of the wires are blocked by an insulating layer to ensure that the thin probe can penetrate deep into the VRFB, and the sensed voltage value comes from a specific local area. Then, stable power is given to the test area, and the voltage difference generated between the two thin wires is measured. The instrument is used to measure the voltage values between the negative terminal pressure plate and the micro-voltage sensor, and the voltage value of the terminal pressure plate is measured at both ends with an external instrument, and the voltage difference formed between the two is the local voltage inside the VRFB value.

### 3.2. Micro-Current Sensor

Micro-current sensors are miniaturized galvanometer probes, which are a set of extended wires. The two probes face the membrane electrode assembly and the graphite flow channel plate of the VRFB, respectively. The sensing area is exposed to the front end, and the rest of the wires are blocked by an insulating layer. The internal current of the VRFB can be guided out to the measuring instrument through the micro-current sensor, as shown in Figure 5.

### 3.3. Micro-Termperature Sensor

We use a resistance-temperature detector, and its electrode type is a serpentine structure, as shown in Figure 6. The temperature-sensing-resistor material is Au because of its stable chemical properties, simple process, and high linearity. The sensing principle, according to the book written by Jewett and Serway [24], mentions the following theories and Equations (1)–(4). In a limited temperature range, when the resistivity of the conductor changes roughly with temperature, it will be expressed according to Equation (1).
(1)$$\rho =\rho 0\left[1+\alpha \left(T-T_{0}\right)\right]$$
where ρ is the resistivity at a certain temperature (T), ρ0 is the resistivity at a reference temperature (T_0_, usually 20 degrees Celsius), and α is the temperature coefficient of resistance. Then, it can be expressed as Formula (2) by Formula (1):(2)$$\alpha =\frac{1}{\rho 0}\frac{d\rho }{\mathrm{dT}}$$
where α is the resistance, and dρ (dρ = ρ − ρ0) is the resistance-change rate between dT (dT = T − T_0_).

It is known from the relationship, Equation (3), of the general metal long straight conductor that the resistance is proportional to the resistivity, that is, the resistance change, Equation (4).
(3)$$R=\rho \frac{L}{A}$$
(4)$$R_{1}=R_{0}\left(1+\alpha_{1}\Delta T\right)$$
where R_0_ is the resistance at T_0_, and the temperature coefficient, $\alpha_{1}$_,_ is the sensitivity of the sensor. Therefore, Formula (4) can be rewritten into Formula (5).
(5)$$\alpha_{1}=\frac{R_{1}-R_{0}}{R_{0}\left(\Delta T\right)}$$

### 3.4. Micro-Termperature Sensor

The type of this sensor is a hot wire micro-flow sensor; the main measurement structure is a resistance heater. A fixed voltage is input to generate a heat source to form a stable temperature field, and then a temperature field is generated above the hot-wire micro-flow sensor. When the fluid flows over the hot-wire micro-flow sensor, the heat will be taken away. The sensing principle is shown in Figure 7. Assuming that the processes are all ideal and fully comply with heat conduction and heat convection, the power provided by the power supply can be regarded as equal as the heat carried away by the fluid. Through the constant temperature circuit design, the flow can be converted into electrical signal output. According to King’s law, the relationship between the heat-loss rate and the fluid-flow rate is as Equation (6) [25].
Q = I^2^ × R = I × V = (A + B × U^n^) (T_s_ – T_0_)(6)
where Q is the electric power supplied by the external power supply; U is the flow rate of the fluid; n is the correlation coefficient between the heat Q and the flow rate U, which is about 0.5 from experiments; T_s_ is the temperature of the hot wire; and T_0_ is the temperature of the inlet fluid. When the flow rate is constant zero, the heat coefficient transferred by the heater, A, is a constant; when the flow rate is non-zero, the heat coefficient of the fluid and the heater is a constant; B is a constant; therefore, Equation (6) can be rewritten into Equation (7).
Q = (A + B × U^0.5^ ΔT)(7)

### 3.5. Reliability Testing of Micro-Temperature Sensors (Temperature Calibration)

Before embedding the VRFB, to make the calibration environment closer to the internal conditions of the battery, we choose the programmable control constant temperature and humidity-testing machine as the benchmark for calibrating the environment. When the VRFB is in operation, the flow channel of the battery will be filled with vanadium electrolyte; thus, the humidity is fixed at 100% during the temperature-calibration process. The temperature-correction range of the micro-temperature sensor is from 30 °C to 70 °C, the interval is 5 °C, and a total of nine signals are captured.

The data-acquisition device we used was a NI PXI 2575 data acquisition unit (Shining, Hsinchu, Taiwan) from National Instruments (NI). The NI PXI 2575 capture device can capture the resistance value of the micro-temperature sensor in real time and use the LabVIEW system designed software and control system for signal processing and analysis. Finally, the analyzed and processed data is output to the computer to draw a calibration curve. The calibration curves of the three micro-temperature sensors are shown in Figure 8. To increase the accuracy, the three micro-temperature sensors are calibrated three times to obtain the average value. After calibration, the curves are all highly linear.

### 3.6. Reliability Test of Micro-Flow Sensor (Flow Calibration)

The flow correction uses the corrosion-resistant liquid diaphragm pump purchased from HOMYTECH, which can provide a continuous and stable to flow. The diaphragm pump is a kind of reciprocating pump. The pump is divided into two parts that do not communicate with each other by elastic film, corrosion-resistant rubber, or elastic metal sheet, which are the areas where the liquid to be transported and the live column exist. The only part in the diaphragm pump that is in contact with the liquid being conveyed is the spherical valve, which enables it to be conveyed in a form that is not attacked by the liquid. The flow range of the liquid diaphragm pump (corrosion-resistant) is 0~400 mL/min, and the flow accuracy is ±2%. Before calibrating the micro-flow sensor, a graphite plate with a single flow channel must be fabricated. The width and depth of the graphite plate is the same as those of the VRFB, which is 2 mm. Then, the micro-flow sensor is embedded in the flow channel. The power supply and micro flow sensor are connected in series with the NI PXI 2575. The power supply will generate a stable temperature field in the form of constant voltage. After the vanadium electrolyte is passed through, the current change value can be measured for flow correction. The flow correction range is from 260 to 300 mL/min, and the measurement is performed once at an interval of 5 mL/min. Additionally, the calibration is performed three times, and the average values are obtained. The calibration curve of the micro flow sensor is shown in Figure 9.

## 4. Results and Discussion

The flexible, multifunctional micro-sensor is embedded in the flow channel. To observe the physical parameters inside the VRFB without affecting its overall performance, the number and pendulum positions of the flexible, multifunctional micro-sensor should be carefully selected. Then, the flexible, multifunction micro-sensor should be fixed to fit it on the runner plate.

After careful evaluation, the three calibrated flexible, multifunction micro-sensors were embedded in the flow channel of the VRFB. The first calibrated flexible, multifunction micro-sensor was embedded upstream of the flow channel. On the other hand, the second flexible, multifunctional micro-sensor was embedded in the middle of the flow channel. Meanwhile, the third flexible, multifunctional micro-sensor was embedded downstream of the flow channel, as shown in Figure 10. Finally, the locking pressure in the endplate was adjusted to lock all sides of the VRFB evenly, allowing stable access to the signals of the flexible, multifunction micro-sensor. Detailed specifications of the flexible, integrated micro-sensor are shown in Table 1.

### 4.1. Performance Tests for Vanadium Redox Flow Battery

After pre-charging the VRFB, the flexible, multifunction micro-sensor and the high-precision acquisition device, NI PXI 2575, are used to perform real-time micro-diagnosis on the interior of the VRFB. Initially, the power supply is used to charge the VRFB of 2 V, where the liquid diaphragm pump is set to flow 280 mL/min, and the NI PXI 2575 set signal is captured once per second. From the beginning to the end of charging, the positive and negative electrolytes of the VRFB changed significantly. After charging the VRFB, it’s the performance is tested with the vanadium redox flow battery test equipment provided by HOMYTECH Global Co., Ltd. (Taoyuan, Taiwan); its voltage performance curve (VI performance curve, as shown in Figure 11) is first measured by gradually decreasing the voltage of 1.6 V by 0.1 V. Then, the current performance curve (IV performance curve, as shown in Figure 12) of the VRFB is measured by gradually decreasing the current of 2.5 A to 0.1 A. Finally, as shown in Figure 13, the maximum power of the VRFB is 0.75 W.

### 4.2. Durability Testing of Flexible, Multifunctional Micro-Sensors

To solve the technical bottleneck, the optimized flexible, multifunction micro-sensor was embedded in a VRFB of real-time micro-monitoring and a 624-h durability test. In addition, the flexible, multifunction micro-sensor was measured at regular intervals. The resistance value of the tester is shown in Figure 14. This data shows that the flexible, multifunction micro-sensor has been tested for 624 h, and it can still be used normally without failure.

### 4.3. Local Voltage Distribution of Vanadium Redox Flow Battery

After using the power supply to set 2 V to charge the VRFB, the flexible, multifunction micro-sensor was used to measure the internal voltage. The result is shown in Figure 15. From the three curves, it can be found that the voltages on the upper, middle, and lower parts gradually increased during the charging period. When the charging reached about 3000 s, the three curves showed an upward trend. In addition, it can be found that the downstream voltage is greater than the midstream voltage and the upstream voltage during the charging period. It is presumed that the upstream should react longer in terms of time, and the upstream is always fed with fresh and unreacted electrolytes. While for the downstream, the entering electrolyte is already partially reacted (charged/discharged already) after passing by the upstream and middle point; thus, it has a shorter reaction time and less severe reaction degree. The midstream voltage curve is located in the center of the flow channel, and the reaction is more severe. Thus, it is less stable than the upstream and downstream voltages.

### 4.4. Local Current Density Distribution of Vanadium Redox Flow Battery

The internal current density was measured with a flexible, multifunction micro-sensor during charging. The result is shown in Figure 16. It can be found that the three curves are highly similar, which means that the current supplied by the power supply is transmitted through the collector plate and is evenly distributed over every part of the flow channel. Further, it can be found from the figure that the current-density curves in the upper, middle, and downstream almost rise or fall simultaneously, which means that the current supplied during charging continuously supplies the vanadium electrolyte for a redox reaction, causing the current-density curve to fluctuate stably up and down.

### 4.5. Local Temperature Distribution of Vanadium Redox Flow Battery

After the VRFB starts to operate, the internal heat energy will be generated due to the continuous redox reaction. Figure 17 shows the temperature-change curve inside the VRFB measured by the flexible, multifunctional micro-sensor. It can be observed that the temperature of the three curves gradually increases with the charging time, and the vanadium electrolyte continuously accumulates heat energy from upstream of the flow channel. Based on the comparison results in the figure, the temperature inside the VRFB is the largest downstream, the second in the middle, and the smallest upstream. Overall, the internal temperature of the VRFB increases by 3~4 °C after charging. The temperature difference among the upstream, midstream, and downstream are about 1~2 °C, while the external ambient temperature is 26 °C during the experimental measurement.

### 4.6. Local Flow Distribution of Vanadium Redox Flow Battery

Figure 18 shows the flow change in the VRFB measured by the flexible, multifunction micro-sensor. Initially, the flow rate set by the liquid diaphragm pump was 280 mL/min. However, the actual flow rate measured in the VRFB was only about 263 mL/min on average. It is speculated that the flow channel was caused by the four-snake shunt. Therefore, the dispersion of vanadium electrolyte would cause the internal flow of the battery to decrease. In addition, it can be seen from the comparison that the flow curves of the flow channels upstream and downstream are quite unstable compared with the flow curve of the middle of the flow channel. It is speculated that because the water inlet and outlet are very close to the VRFB flow channel plate, it is easily affected by the suction and thrust of the pump, resulting in an unstable flow. At the same time, because the flexible, multifunction micro-sensor is embedded in the positive electrode of the VRFB, water will be generated during the charging process; thus, the flow in the flow channel will slightly increase as it goes downstream. The average flow difference among the upstream, middle, and downstream is 1~3 mL/min.

## 5. Conclusions

In this study, the micro-electromechanical system technology was successfully used to integrate micro-voltage, current, temperature, and flow sensors on a PI substrate. The flexible, multifunctional micro-sensor has many advantages, including thin thickness, small structure area, placeable in any position, real-time measurements, and high sensitivity.

The flexible, multifunctional micro-sensor can be embedded on the positive flow channel plate of the battery without affecting the operation of the VRFB. The data, such as voltage, current, temperature, and flow rate inside the VRFB, were successfully read during the charging and discharging process of the VRFB. By knowing the internal operating conditions of the VRFB of operation, improvements can be made in the battery parts and operations in real-time so that the performance and durability of the VRFB can be improved.

## Figures and Tables

**Figure 1 micromachines-13-01193-f001:**
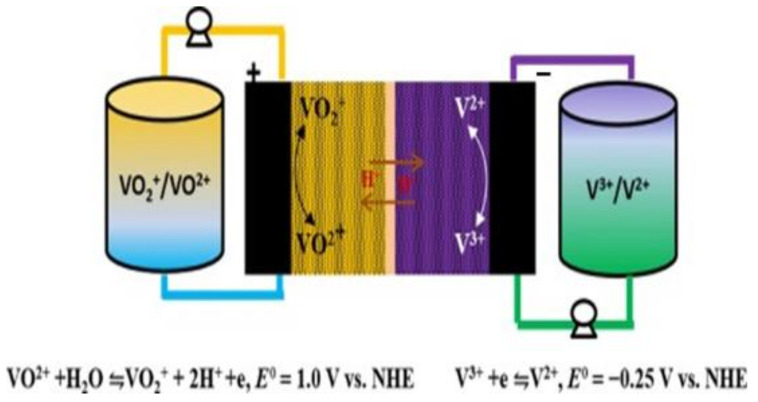
Schematic of vanadium redox flow cell.

**Figure 2 micromachines-13-01193-f002:**
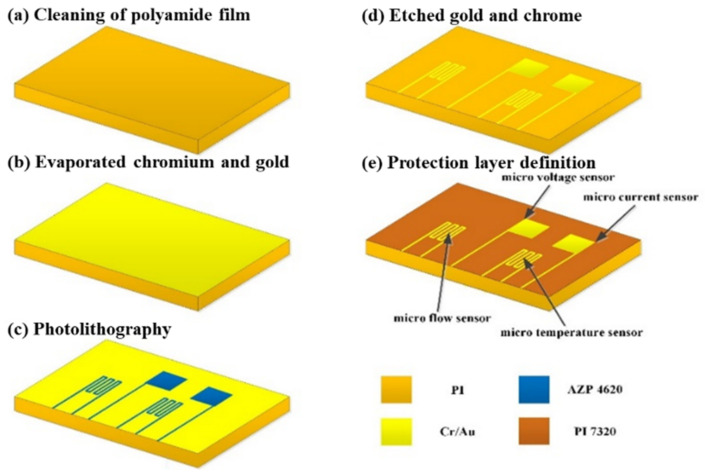
Process diagram of the flexible, multifunctional micro-sensor.

**Figure 3 micromachines-13-01193-f003:**
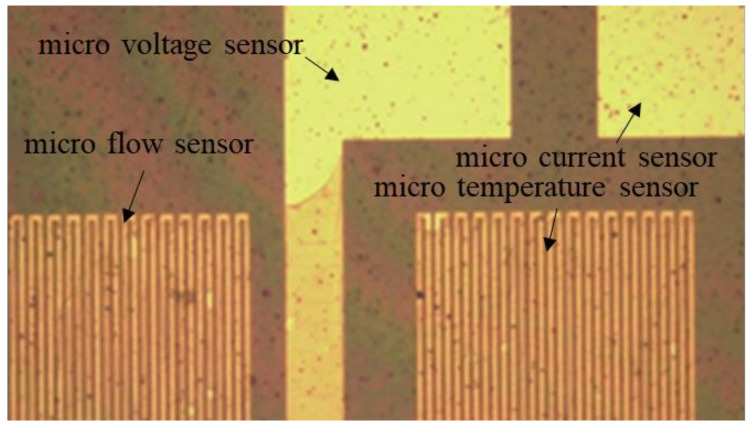
Partial optical microscope view of flexible, multifunctional micro-sensor.

**Figure 4 micromachines-13-01193-f004:**
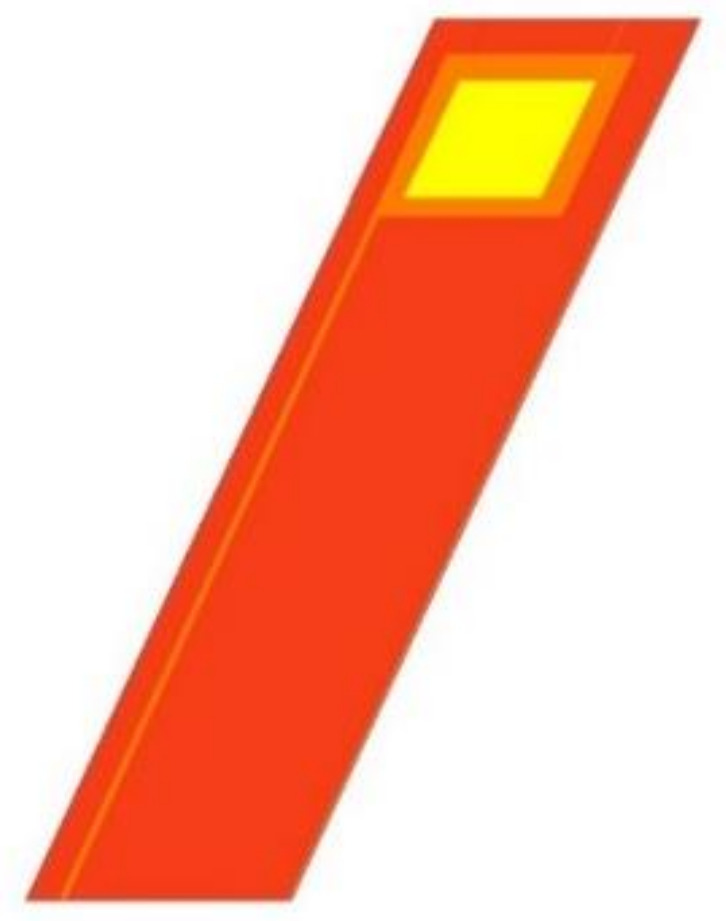
Micro-voltage sensor structure diagram.

**Figure 5 micromachines-13-01193-f005:**
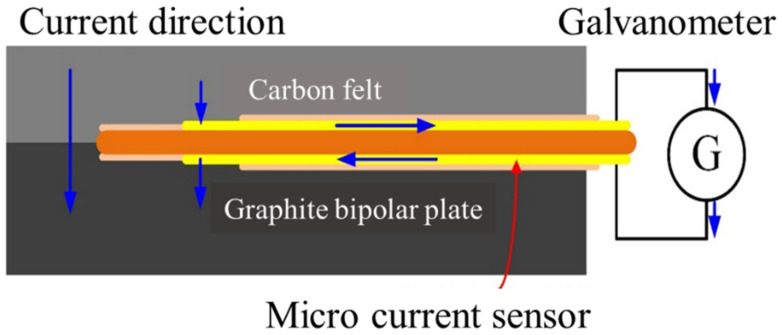
Schematic diagram of the principle of micro-current sensor.

**Figure 6 micromachines-13-01193-f006:**
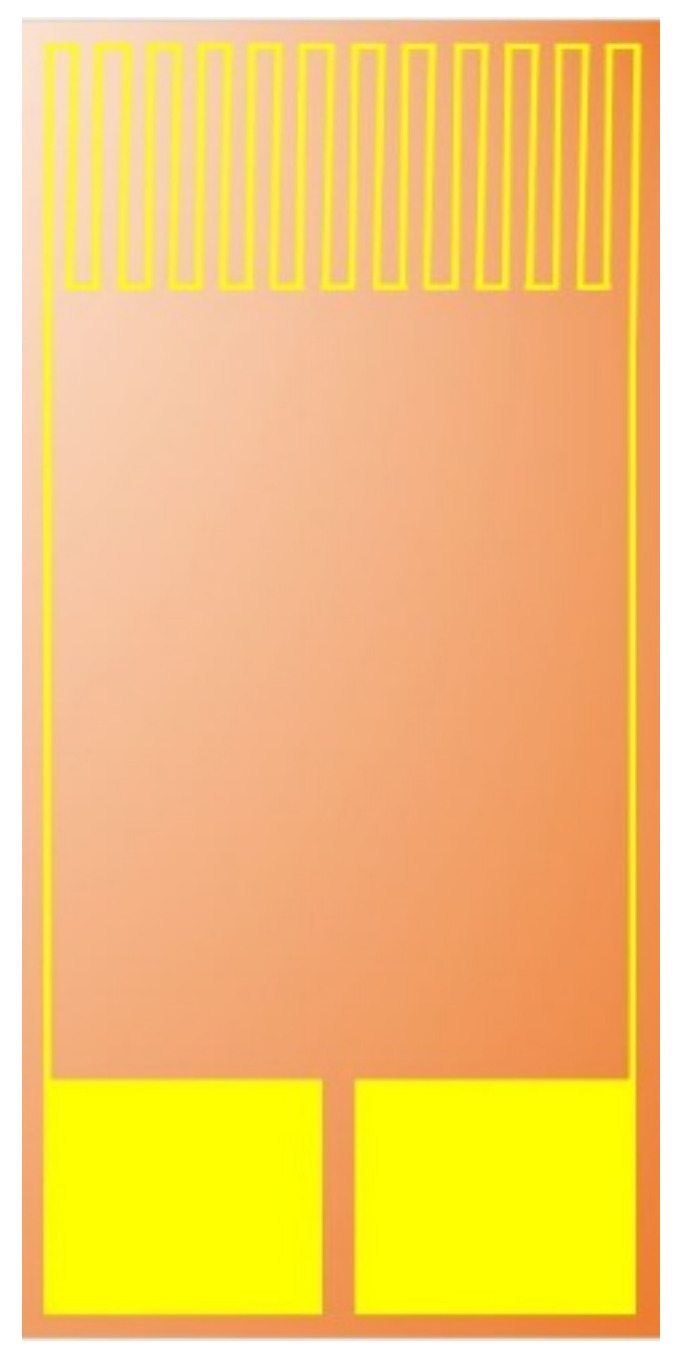
Structure diagram of resistance micro-temperature sensor.

**Figure 7 micromachines-13-01193-f007:**
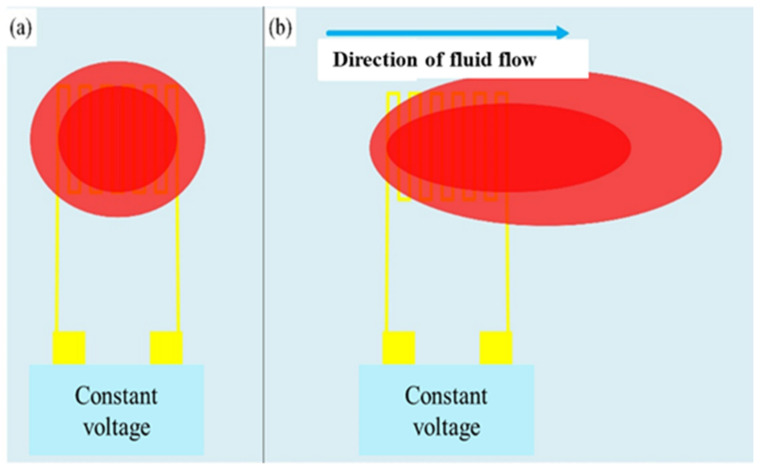
Schematic diagram of hot-wire micro-flow sensor. (**a**) Constant voltage input to generate heat source (red block) to form a stable temperature field; (**b**) Heat is carried away as the fluid flows over the hot wire micro-flow sensor.

**Figure 8 micromachines-13-01193-f008:**
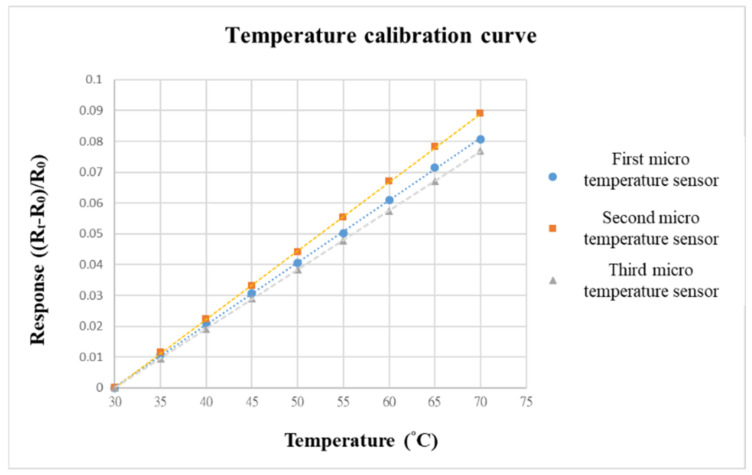
Temperature-calibration curve.

**Figure 9 micromachines-13-01193-f009:**
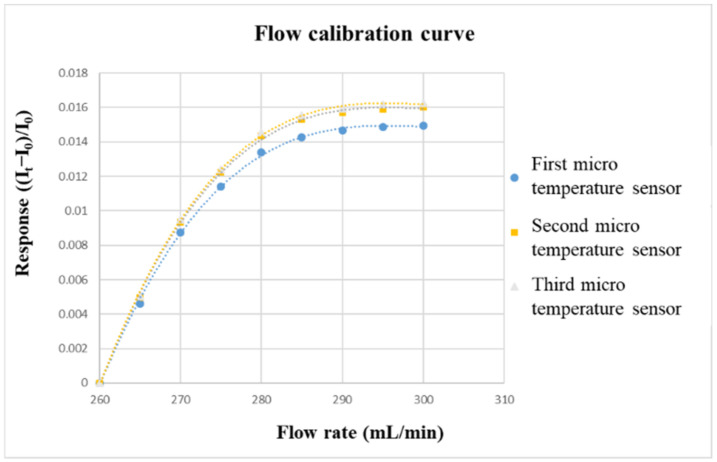
Flow-calibration curve.

**Figure 10 micromachines-13-01193-f010:**
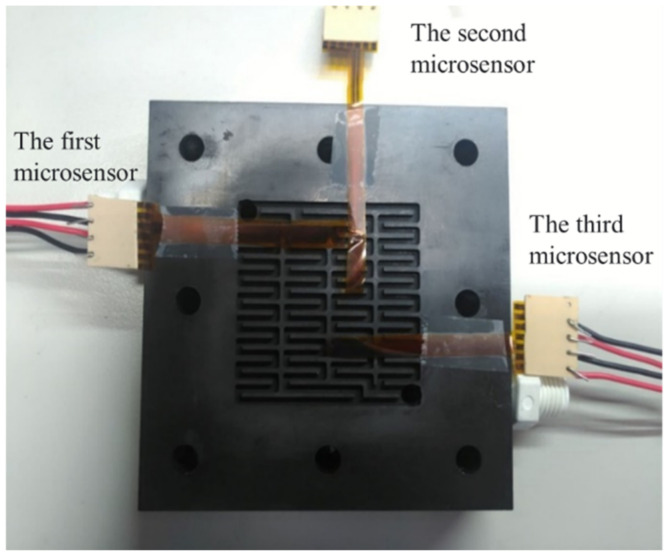
Embedding position of flexible, multifunctional micro-sensor.

**Figure 11 micromachines-13-01193-f011:**
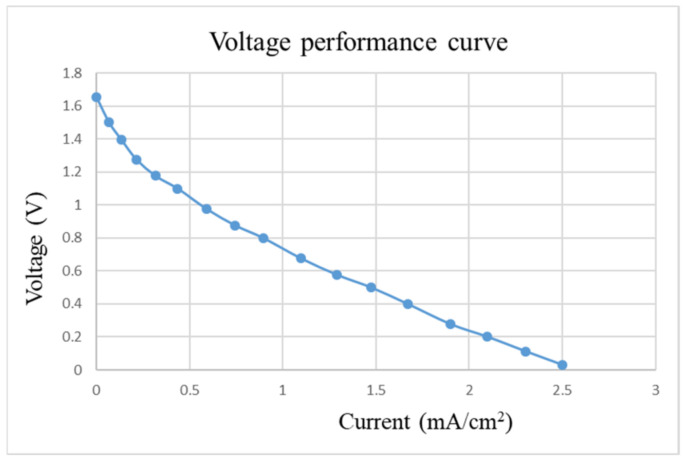
Voltage-performance curve of VRFB.

**Figure 12 micromachines-13-01193-f012:**
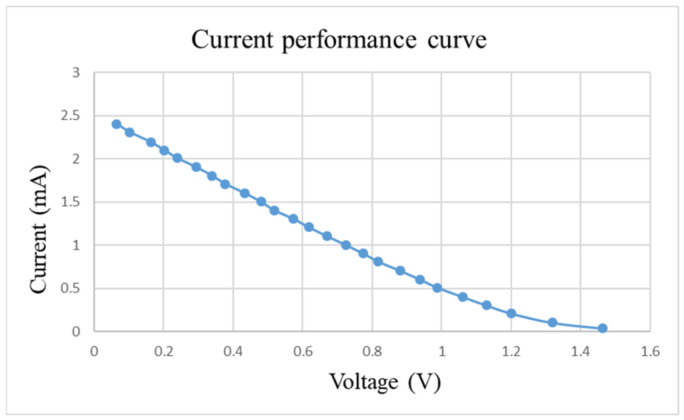
Current-performance curve of VRFB.

**Figure 13 micromachines-13-01193-f013:**
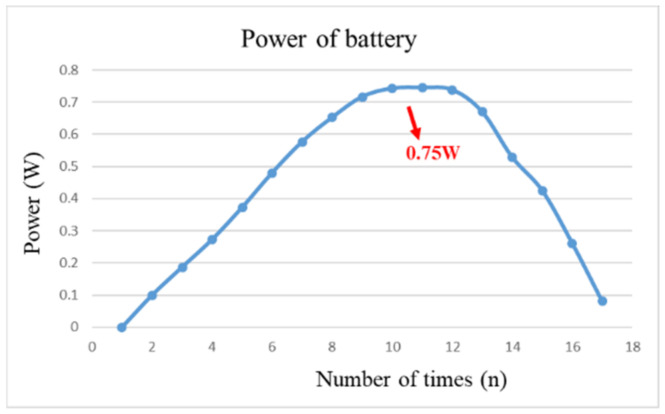
The power curve of VRFB.

**Figure 14 micromachines-13-01193-f014:**
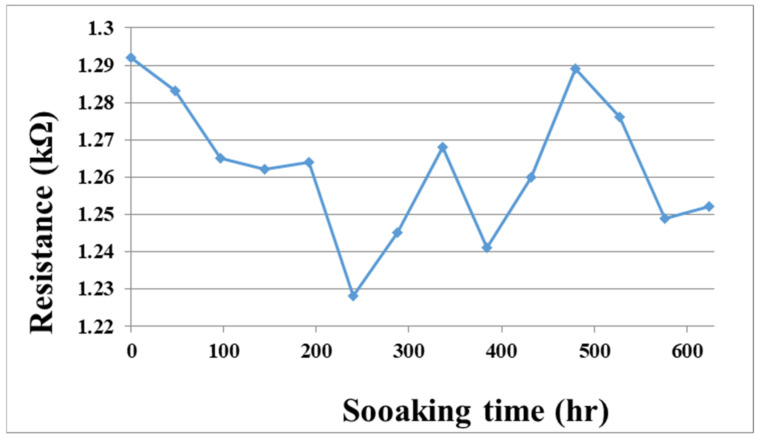
The soaking time and the resistance value of the flexible, multifunctional micro-sensor soaked in vanadium electrolyte.

**Figure 15 micromachines-13-01193-f015:**
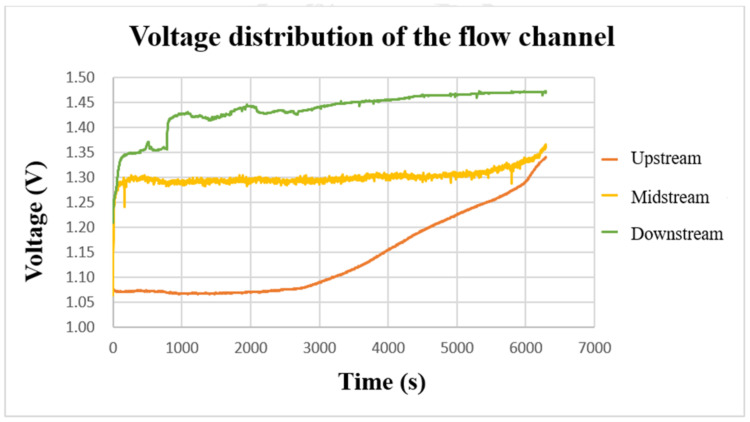
The charging-voltage curve of the flow channel.

**Figure 16 micromachines-13-01193-f016:**
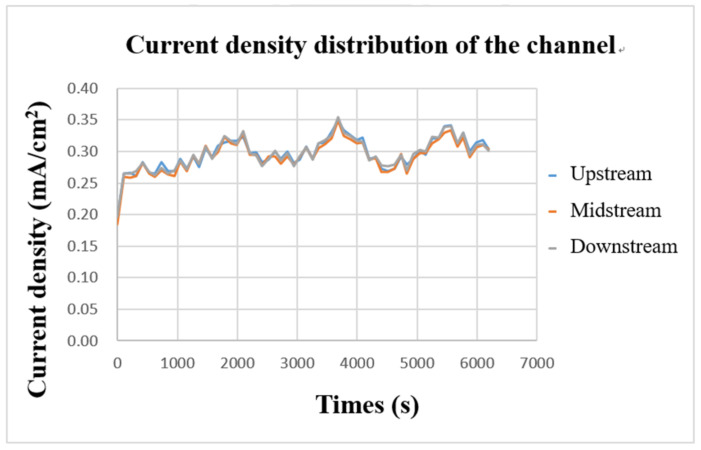
Charge-current-density curve of flow channel.

**Figure 17 micromachines-13-01193-f017:**
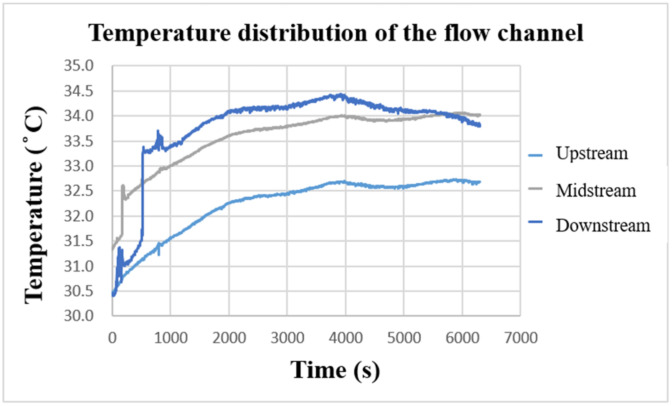
Charge-temperature curve of flow channel.

**Figure 18 micromachines-13-01193-f018:**
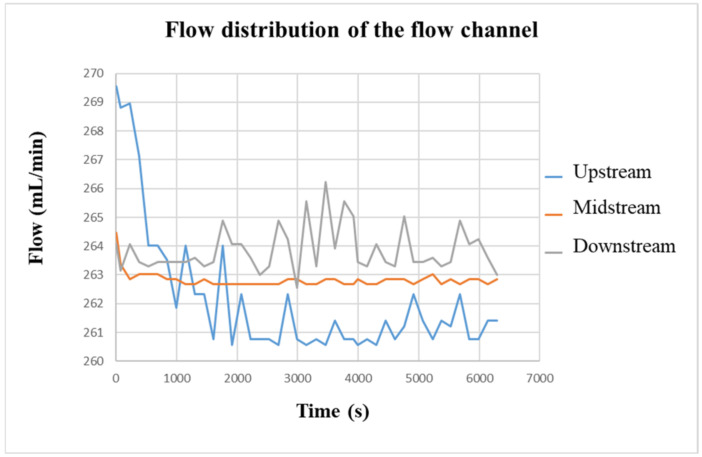
The charging-flow curve of the flow channel.

**Table 1 micromachines-13-01193-t001:** Detailed specifications of the flexible, integrated micro-sensor [16].

	Flexible Micro Temperature Sensor	Flexible Micro Flow Sensor	Flexible Micro Current Sensor	Flexible Micro Voltage Sensor
Size	730μm×600μm×50μm	730μm×600μm×50μm	730μm×600μm×50μm	730μm×600μm×50μm
Operating range	30 ~ 70℃	260mL/min ~ 300mL/min	0~5A	0~5V
Accuracy	≤0.5℃	≤1%	≤0.55mA@1A	≤31μV@1V
Response time	≤1s	≤1s	1μA	1μV
Sensitivity	2.9Ω/℃	0.8mA/(mL/min)	≤1s	≤1s

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
