# Peer review of "Flexible, Multifunctional Micro-Sensor Applied to Internal Measurement and Diagnosis of Vanadium Flow Battery"

_micromachines, 2022, doi:10.3390/mi13081193_

Round 1

Reviewer 1 Report

The focus of this manuscript is to monitor properties of Vanadium (V) - electrolyte in V-Flow Batteries to avoid V precipitation.
Also the use for State of Charge (SOC) Monitoring is suggested, but the sensor is only tested for the positive half cell. For an accurate SOC determination also SOC Monitoring of the negative half cell would be necessary.
Furthermore, the Materials used for the sensor as Au are not suitable for longterm operation in V-Elektrolyte.

Since the scientific background might be not Flow Battery related, relevant references are missing. A very elaborated review concerning SOC Monitoring methods has been published by C. Stolze et al. also half cell Monitoring for SOC determination has been published by S. Ressel et al and T. Struckmann et all and also others.

Chapters 2 und 3 which are supposed to describe manufacturing and function of the sensor was for me not understandable maybe due to a different background. E.g. in 3.1 wires are descrobed and supposed to be shown in Fig. 4. But I could not find wires in Fig. 4. What are the wires made of? Gold? How are they intruduced into the sensor? During the plating process described in chapter 2? A "stable power" (line 208) is described to be supplied to the senor? How is this power generated? 

Also the results and discussion (chapter 4) are difficult to understand. E.g. what is "number of times" in Fig. 13?

To assess whether the sendor is suitable for SOC determination, the results measured should be compared to charge/discharge cycles. E.g. in Fig. 16 the current densities should be related to the operation and SOC of the battery.

I feel not very qualified concerning the language but spelling and Grammar should be overworked: e.g. line 61 singular/plural.

Reviewer 2 Report

In this manuscript, the flexible multifunction micro-sensors are proposed by using MEMS technology for real-time micro-diagnosis in the VRFB. It can simultaneously and locally measure four physical parameters, including voltage, current, temperature, and flow. In addition, it owns extreme environment resistance in the VRFB solution environments. I consider this work to be interesting and involves the key point during the usage of flow battery systems, and thus I suggest a ‘minor revision’.

1. Line 33-40, here I suggest a brief introduction to the classification of the ESS system. VRFB may be the first choice in battery technology, surpassing lithium batteries and fuel cells. But do not forget that the means of energy storage are not only electrochemical but also other physical energy storage methods, such as pumped energy storage and thermal phase change energy storage (10.1016/j.renene.2022.05.129; 10.1016/j.ijheatmasstransfer.2018.09.126). Does VRFB technology have a cost advantage over them? How does technology maturity compare to the two? These are all questions of interest to readers, but there is no relevant introduction in the manuscript.

2. Line 47, Zinc-based and iron-based systems may be cheaper in comparison. [4-8]. But why they are not selected or as commonly used as VRFB? This seems only a half-sentence, but does not completely explain the reason why VRFB is better to be used.

3. Line 69, In terms of the temperature difference in the first flow channel, the downstream and upstream temperature differences decreased with the increase of the reaction time. What is the first flow channel? Are there also second and third flow channels? This issue should be explained more to avoid confusion for the readers.

4. Line 87-89, since it refers to the electrolyte imbalance, asymmetric water crossover, and control of the physical parameters of the VRFB system, some related literature should be added to support this issue (10.1016/j.electacta.2022.139937; 10.1016/j.jpowsour.2022.231640).

5. Line 334-339, how are the VI and IV performance curves obtained? Normally, for the discharging process, a constant current is applied, and then measure the gradual decrease of the voltage. How to realize that the battery is discharged with varied currents and voltages at the same time?

6. Line 361, it can be found that the downstream voltage is greater than the midstream voltage and the upstream voltage during the charging period. It is presumed that the closer the curves are to the downstream, the longer is the reaction time of the vanadium electrolyte and the higher its volt-age.’

I cannot fully agree with this explanation. See Figure 10, I understand that the inlet of the electrolyte should be the 1st sensor (upstream), and the outlet of the electrolyte should be the 3rd sensor (downstream). Do I understand correctly? If this is true, it is not reasonable to say that ‘downstream have a longer reaction time’, because it is the outlet. The upstream should react longer in terms of time, and the upstream is always fed with fresh and unreacted electrolytes. While for the downstream, the entering electrolyte is already partially reacted (charged/discharged already) after passing by the upstream and middle-point, so it has a shorter reaction time and less severe reaction degree.

7. The reference lists are a bit fewer, more related work should be added, especially the inventor Prof. Maria Skyllas-Kazacos group’s work and other simulation-related papers.

8. The description of how the sensors are made is quite long, which should be refined a bit. And more importantly, why the sensors are made by each step should be explained a bit more. Now it seems the readers can only understand how the sensors are made, but why the sensors must be prepared in this specific way is not very clear.

Reviewer 3 Report

Nice work!

L 117 -  needs a verb   "is"

L 346  -  +/- 2%   Standard deviation? Standard error of the mean? Or?

L 389  -  Current (label) is incorrect. It would be just mA, not mA/cm See L                  428

L 391  -  Same problem Power is W, not w/cm2

L 451  -  remove apostrophe

L 458   -  use  -  not   ~

L  18 & 24  -  MEMS   Not sure this device should be designated as a MEMS, sensu stricto. I don't see the mechanical component of it.

A diagram of the placement of the devices in the flow channel region would be helpful.

Is there a patent application for this device?

Round 2

Reviewer 1 Report

Dear Authors,

thank you for answering my questions in a letter. Nevertheless the Description of the production an functioning of the sensor has not been overworked also the results which should reveal the suitability of the sensor for an application in VFB have not been overworked.

Instead, extensive descriptions concerning advantages and functioning of VFB, which also could be found in many reviews, have been added to the manuscript. These additions did not help to understand production, functioning and to convince me about the suitability of your sensor.
